# 'Through the drawings…they are able to tell you straight': Using arts-based methods in violence research in South Africa

Nataly Woollett[1,2☯], Nicola Christofides[1☯], Hannabeth Franchino-Olsen[3☯], Mpho Silima[1☯], Ansie Fouche[4,5☯], Franziska Meinck[1,3,4☯]*

1 School of Public Health, University of the Witwatersrand, Johannesburg, South Africa, 2 Department of Visual Arts, University of Johannesburg, Johannesburg, South Africa, 3 School of Social and Political Sciences, University of Edinburgh, Edinburgh, United Kingdom, 4 School of Health Sciences, North-West University, Vanderbijlpark, South Africa, 5 Department of Social Wellbeing, United Arab Emirates University, A1 Ain, United Arab Emirates

☯ These authors contributed equally to this work.
* Franziska.Meinck@ed.ac.uk

**Data Availability Statement:** The documents have been submitted to the Edinburgh research data

## Abstract

Arts-based methods are underutilized in violence research and may offer improved means of understanding these phenomena; but little is known about their value, especially in low-resource settings. A pilot study using a cross sectional sample was conducted in rural South Africa to determine the feasibility and acceptability of using arts-based methods in research with adults and children, in preparation for a longitudinal multigenerational cohort study on mechanisms that underly the intergenerational transmission of violence. Four arts-based methods were piloted with young adults aged 22–30 years (n = 29), children aged 4–7 years (n = 21) and former caregivers of the young adults aged 40–69 years (n = 11). A sample of qualitative interviews were audio recorded and transcribed (child n = 15, adults n = 19). Three focus group discussions (FGDs) were conducted to understand implementation and lessons learnt with the six interviewers on the study team, none of whom had used these methods in research before. Interviews and FGDs were audio recorded, transcribed and reviewed by the investigative team. Using a rapid analytical approach, our pilot study demonstrated that using arts and play-based methods in multigenerational violence research is feasible and acceptable to participants and interviewers. These methods worked well for nearly all participants regardless of age or ability and offered a comfortable and 'fun' way to engage in weighty conversations. They presented benefits in their capability to facilitate disclosure, expanding understanding, particularly around violence that is often a stigmatizing and sensitive experience. Interviewers required increased capacity and sensitivity in using the methods carefully, to maximize their full potential, and ongoing mentorship was indicated. Our study adds to the burgeoning evidence base of the effectiveness of the use of arts-based methods in health research.

repository and can be found here: https://doi.org/
10.7488/ds/7495.

**Funding:** This research was supported by the
European Research Council (ERC) under the
European Union's Horizon 2020 research and
innovation programme [852787] and the UK
Research and Innovation Global Challenges
Research Fund [ES/S008101/1] to FM as principal
investigator. NW's salary was partially funded by
the ERC. The funders had no role in study design,
data collection and analysis, decision to publish, or
preparation of the manuscript.

**Competing interests:** The authors have declared
that no competing interests exist.

## Introduction

Although arts-based methods in research have been utilized in many disciplines including creative arts therapies, anthropology, sociology, and psychology, their use in health research is more recent [1, 2]. The form of these methods is varied and comprise the visual arts (e.g. drawings, photographs, collage), the performance arts (e.g. drama, film, dance, music) and literary arts (e.g. poetry, storytelling) [3]. Some approaches have been employed in health research across sub-Saharan Africa, particularly in the field of HIV prevention and intervention and primarily using theatre and music [2]. There is burgeoning evidence of the effectiveness of their use in South African health research [4, 5]. Gittings and colleagues (2022) highlight their value in reflecting on and engaging with sensitive topics such as HIV with an adolescent community advisory group over a twelve year period [4], and Honikman and colleagues (2020) support their effectiveness in facilitating dialogue with nurses in primary healthcare settings in an attempt to integrate mental health services more compassionately with patients [5]; but less is known about their utility in violence-related research in a low-resource setting like South Africa. We were interested in exploring their use in preparation for a large, longitudinal cohort study investigating multigenerational transmission of violence in South Africa [6].

Interpersonal violence is pervasive in South Africa and is the second leading cause of loss of disability-adjusted life years [7]. Intimate partner violence is reported by 32% of women in population-based studies [8] and rates of femicide are six times the global average [7]. Violence against children and adolescents is equally rife with national estimates of sexual abuse against girls at 15% and boys at 10% [9]. Given these rates of violence, it is prudent to explore experiences of those exposed to better understand the mechanisms of violence and its transmission across generations [6].

Research that employs non-verbal methods, such as drawing or image making, may offer a principally ethical means of engaging participants in the process of research as these can be more developmentally attuned to the needs of participants, bypassing language and involving participants in dynamic and embodied ways [10]. They may minimise stress inherent in the power dynamic between participants and those who interview them; and through this, voices of participants may be more pronounced and valid, enabling communication [11, 12]. They may also offer a platform that is more easily engaged with where literacy levels may be low, as is the case in South Africa. In the most recent Progress in International Reading Literacy Study (PIRLS), 81% of Grade 4 children (10 year olds) are unable to read for meaning, a statistic that hasn't changed much in the last decade [13].

Concerns that adult data-gathering techniques, that are predominantly grounded in language, may block or distort the perspectives of children have led researchers to seek creative methods to effectively engage children and adolescents in the research process [14]. On account of their participatory nature, arts-based methods are argued to more actively involve children and prevent tokenistic participation [15]. The Convention of the Rights of the Child [16] endorses children's participation and these methods may ensure children's voices are more thoroughly and authentically engaged with and their rights more ethically expressed and facilitated. "Enjoyment, empowerment, and emancipation" are intrinsic to participation and these methods are facilitative of these principles [17]. In addition, a recent qualitative systematic review shows that using arts-based methods in sensitive research with children and adolescents "1) recognize and makes visible previously invisible experiences, acts, voices and histories; 2) nurture change and transformation in the lives of the youth; and 3) allow exploring the more-than-human, more-than-present and less-than-conscious aspects in the lives of youth and children–aspects that traditional study methods might not readily access" [18]. Violence experience can be an 'invisible experience' that is 'less-than-conscious' and research on

violence may benefit specifically from arts-based methods such as drawing, for both child and adult populations.

Another key aspect of studying violence is the compromised mental health that is implicated in trauma exposure of participants. There is evidence that creating imagery in an interpersonal space characterised by kindness, active listening, and empathy has potential to improve mental health [19]. Researchers applying visual and narrative methodologies should be particularly sensitive to the affective impacts of their work, as artistic creation and witnessing can be experienced as an emotive process [20]. The use of image and symbol can support communication for those who struggle to put experience into words, but also for those who are not capable of articulating traumatic exposure [21]. Drawing facilitates verbal reports of 'emotionally loaded' events [22] and resultantly, is a useful tool to assess and elaborate traumatic experiences [23], making drawing especially appropriate as a method in violence research. This may be particularly relevant in South Africa, where trauma exposure in the general population is high [24]. However, importantly, in this process of defamiliarization participants, and indeed interviewers, may be confronted with elements of their lives that they manage to defend from consciousness in their everyday existence [25].

Interviewers thus need capacity to safely manage emotions if they emerge in the process of data collection and the ability to facilitate use of the material, particularly with those who may not be used to the freedoms of utilising them. Especially in low-resource settings, children, adolescents and adults may not have been exposed to more than paper and pencils in their schooling or outside of it. The South African education system has been described as 'essentially dysfunctional' where resources are scarce in the public sector [26]. Importantly, the mental health of interviewers engaged in qualitative research must be recognized as it is both emotionally and interpersonally demanding [27] and may be even more so using arts-based methods. It requires the interviewer to be personally present, open, able to safely manage emotions and probe meaningfully and sensitively. Particularly in violence research, where interviewers are intentionally asking about stigmatizing and sensitive information, often with highly vulnerable participants, training and support needs to be managed and budgeted for appropriately.

Arts-based methods often yield a variety of data, including visual data such as photographs, film, images and drawings that implicate visual ethics, i.e. how to manage data that could identify individuals through its visual nature. The ethical guidelines available for researchers do not always provide sufficient reference to the creation and use of imagery and to the key issues of informed consent, confidentiality and ownership [28]. For those who experience violence, issues of safety and control may be salient and need additional consideration in terms of interaction and what is produced. Ensuring participants feel safe and have a sense of control of the process becomes crucial. This applies to the image created, the interaction around the image with the interviewer, but also relates to space (where and how the image is being made?), materials (are participants familiar with and comfortable using these?) and storage (where are the images kept and can participants take their image with them after the interview if it might bring inadvertent harm to those that live in violent circumstances?).

Notwithstanding these challenges, visual methods, particularly drawing, can be easily utilized in resource-constrained research contexts as there is a simplicity to them; one needs few materials to complete tasks. Drawings are concrete and tangible, inexpensive and easily replicated, and the visual prompt of drawing can lead to storytelling with ease [29]. Drawings often foster personal reflection and insight; the image reveals to the artist something about the experience that might not have been processed and in completing an image and processing it verbally, when one describes the image to another, new information might come to light. This opportunity for reflexivity may be helpful for participants. Also, the methods can help

participants articulate and share events that they may not have talked about before or that are difficult to talk about. Children asked to remember an event through drawing and telling versus telling alone remembered twice more significant detail [22]. Similarly, in a study with older adult eyewitnesses and victims of crime, those who drew a sketch of the event were able to recall significantly more accurate information than those who did not [30].

Drawings have a history of being used in projective testing and assessment as they are seen to reveal underlying psychic information about the artist [31]. As art making is a cognitive behavioural task, it is reasonable to assume that the artwork may divulge something about the artists' thinking and feeling [32]. Projection is commonly regarded as the tendency to externalise aspects of the self and drawings are considered to serve as 'projective containers' [33]. For the purposes of our research, drawings were not used for diagnostic outcomes or assessment, but rather as a platform for personal storytelling.

Due to the existing gaps in the research on the acceptability and feasibility of art-based methods specifically within the South African context and with victims and/or perpetrators of interpersonal violence, a pilot study was conducted. The purpose of the pilot study was firstly, to prepare for a longitudinal multigenerational cohort study on mechanisms that underlie the intergenerational transmission of violence, and secondly, to observe the potential of using four arts-based research methods. This study reports on the experiences of interviewers and participants regarding the feasibility and acceptability of using these methods in violence research with adults and children.

## Materials and methods

### Background on pilot study

A pilot study was undertaken in rural Mpumalanga in South Africa (July–October 2021) [34]. The study comprised a convenience cross sectional sample drawn from residents in the area of young adults aged 22–30 years, children aged 4–7 years, and former caregivers of the young adults. All participants were given a unique identifier number as they enrolled into the study and research data contained no identifying information.

### Interviewer recruitment and selection

Interviewers were recruited through formal hiring processes but came from the same communities participants lived in, with language ability and insight into community that was predicted to lead to high-quality data collection. Disclosure to local interviewers is typically higher than to professional interviewers in violence research [35]. Investment in local staff is crucial in this regard and to increase skills and job security in rural districts. Six interviewers (5 women), ranging in age from 25–48 years were hired, three had high school qualifications and two had completed tertiary studies. All had worked in research studies before, so none were research naïve, but only one had conducted qualitative research. All demonstrated good communication skills and appeared reflective and tolerant. Interviewers did not condone harmful gender norms which were explicitly screened for during the interviewing process.

### Interviewer training and process

There was substantial investment in interviewer training, which is highly recommended in violence research [35]. Virginia Axline, a psychologist and pioneer in the use of play therapy established eight basic principles that guided the facilitation of our research with children (but were generalized to all study participants) and training: (a) build rapport with the child, (b) accept the child unconditionally, (c) establish a sense of permissiveness, (d) reflect the child's

feelings, (e) maintain respect for the child, (f) let the child lead the way, (g) do not hurry the child, and (h) establish limits as needed [36]. Within this, there was emphasis in training on warmth, acceptance, empathy and active listening. These skills were imparted within a trauma-informed frame that prioritized issues of safety, understanding of how trauma impacts individuals, families and communities, and recognition of the importance of self-care in mitigating risk of vicarious trauma. Interviewers practiced using the methods and became familiar with materials and the possibilities of projection and personal expression in image making. There was significant disclosure of interviewer's own childhood maltreatment histories that needed safe management and reflection in training and ongoing mentorship (facilitated by the study team, some of whom are registered mental health clinicians). Disclosure typically occurred when engaged in the experiential practice of the arts-based methods; this was helpful in assessing and raising awareness of how powerful these methods can be in violence research. Some interviewers also disclosed harmful parenting practices in their own lives that required careful attention and discussion to ensure the purpose of the research was fully comprehended and invested in.

## Art and play-based methods utilized

**Quantitative interview.** Quantitative interviews were conducted with 21 young children, 29 young adults, and 11 caregivers. These were interviewer-led, delivered on android tablets, and focused on different types of violence exposure and resilience. Quantitative interviews included arts-and play-based elements, including, for children, a 'feeling faces game' [37], and play-doh house and community plan [38] and, for adults, a Road to Life drawing [39, 40] to allow interviewers to build rapport and collect qualitative data in a playful manner.

The aim of the 'feeling faces game' was to help child participants understand basic feelings of happy, sad, scared and angry in the form of faces. Understanding these feelings comprised being able to name them, recognise them in oneself and in others. Interviewers drew four circles on an A4 sized page and drew simple expressions on each to represent the four feelings. The interviewer then asked the participant to help name the feelings and write the feeling name alongside the face. The interviewer then engaged in a game asking the participant to name the feeling they were demonstrating with their face and body (in the manner of charades) until participants guessed correctly. After completing the guessing game with all four feelings, the interviewer asked the participant to show their 'feeling faces' whilst the interviewer guessed the correct feeling. The game provided a common understanding regarding some of the mental health measures in the questionnaire that incorporate feeling words. The game was also intended to be fun and facilitate rapport between the interviewer and child participant (see S1 File).

The house and community plan using play-doh's purpose was to focus on the child's daily movements from one place to another (see S2 File). With this visual and interactive play-related communication technique, it is possible to identify places and people where a child feels secure and identify areas and situations where they feel threatened. Through this technique, a child is permitted to express positive and negative emotions. The child first draws a plan of their house, then creates figures from playdoh, and uses the figures with the drawing to give details to safe and unsafe places and people in the home and outside of it [38].

The Road to Life tool was utilized to understand positive and negative life experiences pertinent to participants and is an adapted visual representation of the Life History Interview Method [39, 40]. Participants were asked to draw a road on an A4 sized paper with their age, like a timeline, drawn from 0 years to their current age. They were asked to think about all the

good things that had happened in their life and the bad ones and write/draw about the experience next to the numeric age at which it occurred on the line. This method has been used previously with vulnerable participants in South Africa [41].

**Qualitative interview.** Qualitative interviews were conducted with 18 young children, 22 young adults, and 7 caregivers. Children participated in a single qualitative interview that included drawing tasks and lasted approximately an hour. Adults participated in two qualitative interviews, each included drawing tasks. The first session typically lasted one hour and the second 45 minutes. Paper and crayons, pencils etc. were offered for the tasks.

An adaptation of the squiggle drawing game, created by paediatrician and psychoanalyst Donald Winnicott [42, 43] was used to introduce a playful exchange of drawing and guessing. Participants were asked to draw a squiggle on a piece of A4 sized paper with a marker for a few seconds. Then they were asked to use another marker and outline an image in the squiggle that emerged for them. Gentle probing was utilized by interviewers to help participants 'see' something in the squiggle, even if it was a simple shape. The interviewer tried to guess what the image was. The participant then wrote on the page what they identified and interviewers were encouraged to demonstrate appreciation for a task accomplished regardless of drawing or guessing ability. Squiggle drawing facilitated a drawing introduction with the purpose of giving participants opportunity to experience success in drawing and projection, to get creative juices flowing, to engage in a playful encounter and realise there is no right or wrong way to draw (generally everyone can draw a squiggle). The game intentionally sought to alleviate performance anxiety in task completion; validating participants' capacities.

The Kinetic Family Drawing (KFD) [44] was utilized at the beginning of the semi-structured interview to aid the verbal report. Kinetic Family Drawings (KFDs) are projective drawings that provide a useful perspective of a person's social environment, family life and perceptions of family dynamics; they have been used in research globally [45–47] and locally [48]. Children were asked to draw a picture of their family, and adults were asked to draw two pictures, one of their family of origin, and another of their current family. The KFD is useful in understanding changes in family dynamics over time, as well as one's position within families. The participants were asked to explain their drawing once completed using open-ended questions, methods known to increase participation [49]. The frame of the drawing was used to ask and discuss questions of violence and gave some distance to emotionally loaded material through the production and discussion of the artwork [50].

## Consent procedures

Parents gave written informed consent for their participation and written informed consent for their children's participation. Parents were interviewed before children; then children were invited to participate and assented to the research. Participants were interviewed at their homes, either inside or outside in the yard depending on what felt most comfortable to them. Mandated reporting was explained to all participants. All participants consented to their artwork being utilised for research purposes. Imagery was photographed using the camera function on tablets and uploaded to an encrypted server. No identifying names were written on images to ensure confidentiality was maintained. Participants were asked what they wanted to do with their artwork when the interview was completed and could either destroy it, leave it with the interviewer, or take it with them. Participants were asked if any of the imagery may place them in harm's way if they took it with them. In these cases of perceived threat, participants were encouraged to leave their artwork or destroy it. Participants in focus group discussions completed written consent for participation.

**Ethical considerations.** The study adhered throughout to the WHO ethical guidance on IPV research [51], UNICEF guidelines on ethical research involving children [52] and guidelines for the prevention and management of vicarious trauma among researchers of IPV [53]. Full background checks were conducted for all interviewers. A thorough distress protocol was utilised with a trauma-informed approach to research engagement and participant interaction [54]. Strategies used to confirm participants were provided with a safe environment included asking where to conduct the interview, whether it was safe to be there, engaging in play with the children, and gathering data via arts and play-based methods. As a reimbursement for their time, adult participants (young adults; caregivers) received a R50.00 ($3) grocery voucher for each interview they participated in, and children received a gift pack (valued at $5) in their initial interview (quantitative) that comprised an activity book, washcloth, and soap, and were later given a small contribution (stickers; a pen) if they engaged in multiple interviews. Participants also received refreshments (a juice box and biscuits).

The study employed a full-time social worker to attend directly to any mandated reporting requirements and further support participants and their families. Participants were made aware of this before engaging in interviews.

Ethical approval was granted by the University of Edinburgh School of Social and Political Science Research Ethics Committee (264227), the University of the Witwatersrand Human Research Ethics Committee (M190949) and North-West University Health Research Ethics Committee (NWU-00329-20-A1). Further ethical approval was granted by the Mpumalanga Department of Health (MP-202012-003). Full ethical approval by all ethics boards took 26 months (see S3 File). In addition, local area government gave permission to research being undertaken after discussion and presentation of the study to the chief, indunas and ward councillors.

Additional information regarding the ethical, cultural, and scientific considerations specific to inclusivity in global research is included in the S4 File.

## Data collection

A portion of qualitative interviews were audio recorded and transcribed (with children, n = 15 and young adults, n = 19). Interviewers completed fieldnotes after these to assess distress and feasibility of methods used. In addition, three focus group discussions (FGDs) were undertaken with interviewers following pilot study completion to understand implementation and reflect on interviewer perceptions, experiences and lessons learned. The FGD guides were developed by the investigative team (five members) and comprised an intensive, team-based approach that helped strengthen insight and validity of findings. FGDs involved all 6 interviewers over a period of 4 months. These discussions were guided in person by the project manager, who is a qualitative researcher, and were supported online in real time by the investigative team. Supplementary questions were asked by investigators during the discussions that took roughly 2 hours, were audio recorded and transcribed. Observation notes were undertaken when investigators were in the field and the investigative team met weekly to discuss implementation, with meeting notes recorded, and reflect on data jointly as it was gathered and submitted (see data repository for consent forms, FGD guides and framework for quotes).

## Data analysis

First, four researchers reviewed and assessed at least three transcripts each of interviewer-led qualitative interviews, to appraise pilot processes and application, giving feedback to interviewers on utilization of methods, interaction with participants and quality of data gathered. Researchers discussed reflections of the transcripts reviewed (that were grouped into themes

by the first author) and together with observations from the field and team meeting discussion over a 5 month period, triangulation primed inclusive discussion guides for the FGDs that followed [55] as well as themes that were identified in participant transcripts. Deliberations on central findings after every FGD was concluded with the broader investigative team ensured nuanced questions were incorporated in the next round of discussions, guaranteeing all areas of investigation were being grasped and studied. A rapid analytic approach [56, 57] was used to analyse data. This swift and iterative approach to data collection and analysis was undertaken to understand the pilot study, including the acceptability, feasibility and appropriateness of the study methods and materials, and chosen for its capacity to produce targeted research in a timely way to inform the main study tools and implementation that was forthcoming [58]. Study investigators who analysed the data were embedded in the study with operational understanding of the context and methods, offering insight to the complexity of the study implementation. The analysis was minimally interpretive with the first author establishing an inventory of the data contents or main domains, derived from the interview and discussion guides, condensing and consolidating the data into summaries [59]. Three researchers read all FGD transcripts, underscoring significant quotes that matched the inventory generated, and clarified or confirmed summarized significant findings collectively.

## Results

### Benefits of methods used

There were no refusals in using the methods and all participants were able to complete a drawing. The process of drawing was more important than the perceived value of the end product and all could participate successfully, regardless of drawing ability. For child participants in particular, drawings were easily completed and play methods experienced as fun. However, especially with older caregivers, who might have been illiterate, there was a sense of fear in not being able to write or draw and more sensitivity and encouragement was required on the part of the interviewer. In these situations, interviewers typically modelled how to draw and gave the participant the opportunity to try in a non-judgemental encounter.

There was novelty in using these methods, both on the part of participants but also for the interviewers.

*I think the drawings, it like uhm, those drawings have messages that the children wouldn't be able to tell you straight, but through the drawings the child is able to tell you through the drawing (P5, FGD3).*

*Yes, I was asking myself, why are we doing this? Why are we doing the playdoh? And I kept asking myself will I be able to interview a child? A child? A four-year-old? And after a while after I was doing an interview I realized, "this is very easy, a child can communicate with you" (P4, FGD3).*

Frequently, participants (and interviewers), were surprised at what could be elicited from the images produced and participants also seemed to feel a sense of achievement or accomplishment through their completion of tasks. Drawings seemed to facilitate discussion, particularly regarding violence experience.

*Children carry a lot of information regarding violence and the only way you can find out is how you ask them, how you engage them, through the drawings they help you get that violence out the child. I believe they give you first hand, raw information (P2, FGD3).*

Disclosure appeared to occur effortlessly with participants regardless of the sensitivity of the material shared; often participants spoke spontaneously about the violence they experienced without being asked directly. Through this participants shared weighty personal experience that seemed to lead to meaning making.

**Interviewer.** Thank you for taking your time to draw this pictures, can you tell me what is happening on this picture?

**Respondent.** Person 1 is me I was ducking a stick, person 2 is my father who was punishing me, person 3 is my mother she loved going to the farm to plough most of the time when my father was out drinking and not home. Person 4 is my grandfather he was still alive, he was a carpenter. My grandmother was a farmer and these are my siblings. My female siblings loved playing outside and my younger brother loved herding the goat.

**Interviewer.** You told me that there was a time that person 2 [father] would fetch you from school, can you tell me what happened with your studies at school?

**Respondent.** It was embarrassing even though he would hide the stick, I felt like I was not important to be taken from school while others are busy learning. But things started to be better as I grew up, my father was no longer fetching me at school but he would want me to go herd the cattle when I come back from school. I finally decided to quit school because the situation at home was hard and no one was working then I decided to quit school and look for a job to help out at home and buy myself clothes. That is the reason why I also named my drawing 'Tiyiselani' [perseverance]

**Interviewer.** Oh that's is why you named your drawing Tiyiselani?

**Respondent.** Yes because the situation at home was hard. Sometimes when our parents fight we cry because he [father] was beating our mother. It was not easy, we had to be strong (Young male adult participant, P1).

The ability to reflect on relationships in childhood and in present family constellations through the KFDs across time in their lives may have been particularly helpful in recognising intergenerational patterns of behaviour and interpersonal relationship styles that transmit over time.

Part of the strength of the arts-based methods is their ability to help participants remember past events and reflect on them in the present.

I think it was the Road to Life, because that one has certain stages, age, years where one is forced to remember. Because if you asked me what happened at the age of 5, here based on what I wrote, talking about it, getting it out, maybe it's something you didn't want to share but now it's out there we must talk about it, it brought out some things (P2, FGD3).

This also seemed to lead participants to reflect meaningfully on their images and gain insight from them.

**Interviewer.** What stops families from having fun?

**Respondent.** It's poverty, when you don't have money you won't be able to have fun, as from both pictures people are not employed so it made it difficult to have fun as a family because we don't have money (Young female adult, P2)

They [adults] were normally saying these questions and drawings helped them a lot and they started to realize something that they didn't do in their life (P1, FGD1).

In addition, this reflection lead to rich discussion during interviews.

**Interviewer.** If you could change anything in either of these pictures, what would that be?

**Respondent.** I would call for a meeting and tell them [family elders] that since I have grown up, I would also like to be heard in this family and tell them that the way of living in this family it is not right and suggest a better way of living.

**Interviewer.** What would you change?

**Respondent.** We [all children] would go to school, they [adults] would stop the beating and they would discipline the children by a way of sitting them down and talking to them when they did something wrong, that is what I would change (Young male adult, P3)

Interviewers revealed the powerful potential of using these ways of working with participants; a realisation that came from their experiential use of the methods from training. It also highlighted the need for these to be used with sensitivity and care.

**Respondent.** The time we were at the training something came out from me that I didn't know I was bottling inside, but then the time we did the drawings it came out as though someone was asking me even though I was just having the drawings.

**Interviewer.** So, the drawings specifically stuck out to you because it made you think about your own personal experiences? And how was that process for you when you were now starting to remember things from your past from this drawing?

**Respondent.** I'm surprised at how it came out because I've never been open to anyone like that. I'm glad it happened because I feel better now because I'm better now because I know what I was bottling inside but then since I let it out. I think ya, that's the most part I loved about qualitative interviewing (P3, FGD3).

The arts-based methods might have been particularly helpful in regulating emotion for some participants.

Ya, like I remember this child. He is six years old so he was so excited [difficult to manage] and I didn't know that by doing that [playing] I will get what I'm looking for and he will answer all the questions that I am asking and only to find that at the end that the person is relaxed and I can start my questions (P4, FGD3).

## Challenges in using the methods

Initially difficult for interviewers to understand and engage in, use of the methods required mentorship and practice throughout fieldwork.

*I remember the squiggle drawing. I was like so what is going on here, why are we doing the squiggle, how is it going to help and the playdoh 'what is this for*?!' *This thing is so difficult, only to find that when you are in the communities doing the interviews it helps you so much. Because you have to learn that the squiggle drawing has to help your participant to relax, getting to know each other, gaining the trust from that person (P4, FGD3).*

The 'joint attention' required to engage deeply, and the understanding of projection was limited for some of the participants but was also difficult to master for some of the interviewers.

*Like on the squiggle drawing, maybe you would ask the participant "what do you see here?" then the participant will tell you something you don't even see. But then the participant says that they see that thing, you can't say 'no you are lying'. You have to say that you also see. . .- most of the participants I had, would tell you something I don't see there (P5, FGD3).*

Some interviewers were shocked at what could be elicited through the methods and on account of this might not have probed as deeply as possible, e.g. exploring children's fantasies that show up easily in non-verbal ways.

*For me, when I was doing the house drawing with the playdoh. The child stays in the house with the whole family. He drew the house and then another house, a small room outside the drawing. Its only when I probed to ask who is the person staying in the outside house, that it was the mom's sister. He kicked that one out of the house in his mind and drew her separate because he hates that person. So, it was shocking (P2, FGD3).*

In addition, for some interviewers, engaging in the use of these methods required practice and being playful or expressive couldn't be assumed to be a 'common skill'.

*On my side the training helped me more because I wasn't able to show someone my angry and sad face, it was difficult but then through the training I have learnt and I was enjoying when I was doing it with the child (P5, FGD3).*

We did find that some participants, typically those less developmentally advanced, struggled to engage, which could be the result of behaviours not being managed well in the interview or being easily emotionally triggered by the tasks required.

*Mine, she was challenging. Sitting here, going to sit on another chair, asking about the tablet, she wants the marker. She told me that she's tired. So I had to take maybe one biscuit and give it to her so that she could concentrate and come back to me but as soon as she was finished she was up again, so then I had to keep on promising her that this will finish (P1, FGD2).*

This 'difficult behaviour' was pronounced in the house plan activity with the squiggle drawing and KFD typically being easier to complete independently.

## Discussion

Our pilot study found that using arts and play-based methods in multigenerational violence research is feasible and acceptable to participants and interviewers. These methods worked well for nearly all participants regardless of age or ability. They can enhance what researchers (and participants) can discover through their capacity to facilitate disclosure, particularly around stigmatizing and sensitive experiences like violence [21]. Perhaps this is due to the increased safety these methods provide in their ability to facilitate expression and allow participants personal distance from the content of violence they are sharing with interviewers [21].

Creative arts therapies are known to support mental and physical wellness, improving the ability to communicate experience, express feelings and feel safe [60], which may position these practices well for their inclusion in equitable research, especially for those who may

experience conventional methods as 'unsafe' or constrictive. Creative arts therapies have been proven effective for many mental health problems, but especially for treatment of trauma and complex PTSD [19, 61, 62]; a potential outcome of violence exposure. These methods enhance memory recall, enabling memories to be processed and for the 'artist' to have control over this process [63], an important aspect to consider in violence research where participants may have experienced significant lack of control. The creative product may function as a container for affect, promoting emotional regulation [64]. Words and meaning can be ascribed to memories manifest in tangible form for participants through the product; they can be arranged and consolidated to produce coherent narratives and eventually to integrate experience [65]. Importantly, this can also encourage violence being named as such, validating the feelings or thoughts associated with this exposure [66, 67]. Being listened to when communicating personal history to another (i.e. participants talking about themselves through their drawings without interviewer interpretation) can reduce the isolation many feel as a result of experiences marked by stigma, improving self-worth and agency which have knock on positive effects for mental health [68].

Interestingly, a bulk of creative arts therapy research with patients who have experienced trauma is supported with neuroscientific findings that recognize the importance of arts-based strategies that do not rely solely on language for processing [69–71]. The neuroscience behind imagination has shown that when people visualise something and remember it, the activity shown in their brain resembles that associated with the real experience [72], suggesting that our imagination has a real and tangible impact upon how we feel. This means that imagined phenomena are not just a 'figment of our imagination'; they are part of our embodied experience and to manage them we may need our body's investment in processing; something art and play-based approaches facilitate in clinical practice, but perhaps in research too.

Engagement in art and play-based methods may hold therapeutic value for research participants. The mechanisms that underlie mental health change in the creative arts therapies are understudied but emerging evidence highlights embodiment (physicality of the arts, body awareness, engaging mind-body connection), concretization (the image or product facilitate verbal reflection and self-knowledge), symbolism and metaphor (unconscious processing and externalization, self-expression), enhanced self-efficacy and motivation [60]. Similarly, the mechanisms for change in play therapy include facilitating communication (self-expression, teaching), fostering emotional wellness (counterconditioning fears, stress inoculation, catharsis), enhancing social relationships (attachment, social competence, empathy), and increasing personal strengths (creative problem-solving, resiliency, self-regulation, self-esteem) [37, 73]. There is possibility that these positive effects could be experienced through participation in research as well. Through the process of artmaking, play, and joint attention, participants may increase their ability to become more self-reflexive and self-aware about the issues being investigated. Through reflection one is better able to gain a perspective and when this is shared within a confidential, supportive and attentive interpersonal interaction, in some instances, the action has transformative potential. This is particularly the case with stories that may not have been told or heard before, which is often the case in violence research. Our study supports other local research with vulnerable violence exposed participants and their perceived benefit of inclusion, having an opportunity to share adverse experiences with others in a 'safe space' [74].

It is possible that the most at-risk participants had more difficulty focusing and concentrating, regardless of age. Research demonstrates that children in South Africa are significantly exposed to violence during childhood which places them at risk for trauma [75]. Often traumatized children may present with symptoms of attention deficit disorder (ADHD) that is more likely to be complex PTSD related or that these two phenomena are highly interrelated [76]. Dysregulated affect, hyperactivity, and irritability are difficult symptoms to manage in a

research interview and may require additional expertise and resources. It also has implications for any violence-related interviewing of children and adolescents in the South African context, or anywhere that violence against children and youth is highly prevalent.

We found that children as young as 4 years old could participate in our pilot study [34]. With very young children, researchers have highlighted that verifying the information they provide may be difficult, and that data elicited may not be a genuine reflection of the way they understand the world [77]. Drawing has been advocated as a possible pathway to true representation of children's thoughts and may elicit communication in a way that best exemplifies the way they see the world [78]. A marked strength of drawing is its direct and immediate aptitude to reflect back to its maker the thoughts derived through its making. Perhaps this explains young children's seemingly natural interest and gravitation towards drawing. It is instantaneously all-inclusive and interactive in a manner that writing and speaking are not. Children's felt sense of understanding of their contexts might easily be given expression in drawings. But, for children to be afforded the opportunity to engage in the complexity of their drawings and the time required for such, there needs to be emphasis of these necessities in the training and research provided. Drawing in our study led to improved mutual understanding between child participant and adult interviewer. Disclosure also led to increased referrals for some vulnerable children, another reality to be managed purposely. Although no child participants showed obvious levels of distress by talking about violence or drawing pictures, the circumstances in which children lived and the experiences they had were distressing. Interviewers were trained to manage disclosures appropriately and provide empathic support to participants, whether children or adults, through processes of mandated reporting. Having a full time social worker on the project who intervened in these cases and ensured they were effectively managed was a source of support to participants as well as interviewers. This is particularly important in contexts where child protection and social services may be ineffective or unobtainable.

In drawing or creating any arts-based product, there is an image or artwork that is produced. Used in research, this creates a triangular relationship between participant, interviewer and image. The participant and the interviewer join their gaze and attention towards the image which gives voice to the process and experience of the participant. Instead of looking *at* the participant, the interviewer looks *with*, facilitating a more comfortable dialogue and storytelling perspective. This is significant in cultures where making direct eye contact as children with adults might be perceived as disrespectful or disobedient [79]. The image may bridge these cultural influences and may create safety in directing shared gaze towards the third entity in the triangle. The idea of 'joint gaze' is one grounded in developmental psychology and art therapy where a participant and adult join together in joining their gaze and by extension, their attention [80]. The capacity for joint attention cannot be taken for granted and is a skill that needs mentorship and development in interviewers. It is also a skill more challenging to utilize with those who may have attachment difficulties [81, 82], and the intimate partner violence environment tends to put caregivers and their children at risk for perpetuating insecure attachment relationships [83]. The implication is that the participant requires the underlying cognitive and emotional skills [84] of joint attention to be able to look, think and reflect on their own position and the position of their image during the interview. Thinking about what the image may reflect or mirror back [85] may need to be facilitated and supported by the interviewer within the triangular relationship and may be too sophisticated a skill to master in some circumstances.

Interviewers seeking this data need adequate practice with consistent training regarding the power and potential of sharing imagery interpersonally around violence. Values clarification is important in training, both regarding violence, the use of imagination, and art and play as methods of data collection. Reflexivity is a vital ethical issue to consider in using non-verbal

approaches and should be consistently engaged with. The visual image is a commanding medium of communication, but visual ethics need intentional guidance and oversight. Participants must consent to their image being used in research and more broadly being shared with others outside of the research interview. Interviewers should ensure that no identifying information is held within an image that can reveal the identity of the participant, but also their family members. This raises tensions between revealing and concealing the contents of visual images and who has 'the right' to claim ownership of images and show them to others [25].

The visual images that researcher's study and engage with form part of a wider narrative that directs their framing, and when we ask participants to create something visual, our understanding of these images is often grounded not just on the image itself but the consequent elicitation interview. The interviewer should be encouraging of creativity and respectful of what it yields for it to be effective. Images themselves are also open to interpretation and reinterpretation, both from those who create them, but also those who get to look at them and as Hall maintains, images rarely contain a singular or 'true meaning' [86] but invite dialogue and meaning making opportunities. Interviewers must be alert to their own bias in interpreting images without getting proper feedback from participants on what they themselves attribute to their imagery and storytelling about them. In addition, power and positionality, both between participant and interviewer, but also between interviewer and image, need a frame that is tolerant and without judgement; one that is imbued with interest and curiosity. This is difficult to achieve without practice. It is helpful for interviewers to be informed about participant developmental capabilities in drawing and play, i.e., being privy to the drawing abilities of children and adults across development [87], recognising that there may be delays in these abilities and just because a child can't 'draw like a 5 year old' doesn't mean that their image can't tell their story. Lastly, the costs of the emotional involvement (of both participant and interviewer) in reliving their negative experiences during interviews and through drawing needs to be dealt with in research [88]. The investigative team and line managers were available and encouraged interviewers to discuss interactions that were challenging or triggering for interviewers, purposively creating a research culture sensitive to vicarious trauma possibilities and being responsive to it.

In our review of the literature, we could not find studies from South Africa that investigated the use of arts-based methods utilized by fieldworkers/interviewers conducting violence research, i.e. not utilized as an intervention per se and not applied by specialist therapists. This indicates an opportunity to explore in future research to assess the utility of these methods in cross sectional studies on violence.

## Limitations

Rapid data analysis helped identify the context specific issues that needed to be addressed and understood (both on the part of participants but also interviewers) related to the data collection tools utilized for the pilot. However, rapid research designs tend to use small sample sizes which complicates generalizability of findings. Also, this analysis strategy did not benefit from a deeper thematic analysis of data.

Although interviewers where trained and mentored in the methods used, there was a limitation in deepening interviews and facilitating more probing around significant participant responses. As a result, the main study employed more experienced qualitative researchers. Participants may also have been biased through social desirability.

## Conclusion

Our pilot study revealed that arts-based methods can lead to rich data around violence experience in children and adults alike and can be successfully implemented in a low resource setting

in rural South Africa. These methods, used in violence research, may offer participants benefits through active engagement, joint attention, memory recall, personal expression and integration of narratives. It may also be beneficial to have an opportunity to disclose personal adversity through these means. Interviewers require increased capacity and sensitivity in using the methods carefully, to maximize their full potential, and ongoing mentorship is indicated. These approaches are underutilized in health research and may offer improved means of understanding violence whilst ethically engaging participants.

## Supporting information

**S1 File. Feeling faces game.**
(DOCX)

**S2 File. House community plan.**
(DOCX)

**S3 File. Ethical approvals.**
(PDF)

**S4 File. STROBE checklist.**
(DOCX)

**S5 File. PLOS questionnaire.**
(DOCX)

## Acknowledgments

We would like to acknowledge our participants and dedicated interviewers.

## Author Contributions

**Conceptualization:** Nataly Woollett, Nicola Christofides, Ansie Fouche, Franziska Meinck.

**Data curation:** Nataly Woollett, Nicola Christofides, Hannabeth Franchino-Olsen, Ansie Fouche, Franziska Meinck.

**Formal analysis:** Nataly Woollett, Mpho Silima.

**Funding acquisition:** Franziska Meinck.

**Investigation:** Nataly Woollett, Nicola Christofides, Hannabeth Franchino-Olsen, Mpho Silima, Ansie Fouche, Franziska Meinck.

**Methodology:** Nataly Woollett.

**Project administration:** Nicola Christofides, Hannabeth Franchino-Olsen, Mpho Silima, Franziska Meinck.

**Resources:** Nicola Christofides, Hannabeth Franchino-Olsen, Franziska Meinck.

**Software:** Hannabeth Franchino-Olsen.

**Supervision:** Nataly Woollett, Nicola Christofides, Franziska Meinck.

**Validation:** Mpho Silima.

**Writing – original draft:** Nataly Woollett.

**Writing – review & editing:** Nataly Woollett, Nicola Christofides, Hannabeth Franchino-Olsen, Mpho Silima, Ansie Fouche, Franziska Meinck.

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
