## [Decision Letter · Decision Letter 0]

7 Jun 2023

PGPH-D-23-00851

‘Through the drawings…they are able to tell you straight’ : using arts-based methods in violence research in South Africa

Dear Dr. Woollett,

Thank you for submitting your manuscript to PLOS Global Public Health. After careful consideration, we feel that it has merit but does not fully meet PLOS Global Public Health’s publication criteria as it currently stands. Therefore, we invite you to submit a revised version of the manuscript that addresses the points raised during the review process.

This is a fascinating and innovative study. Congratulations!

Please address the comments/critiques raised during peer review. In the introduction, it is important to include one paragraph describing the context of domestic violence in South Africa (prevalence, major trends, etc.). In your discussion section, please compare your results with similar studies addressing domestic violence in SA and/or studies that utilized arts to understand better the impact of domestic violence on people’s lives.

We look forward to receiving your revised manuscript.

Kind regards,

Monica Malta

Academic Editor

Journal Requirements:

2. Please send a completed 'Competing Interests' statement, including any COIs declared by your co-authors. If you have no competing interests to declare, please state "The authors have declared that no competing interests exist". Otherwise please declare all competing interests beginning with the statement "I have read the journal's policy and the authors of this manuscript have the following competing interests:"

3. We noticed that you used "unpublished data" in the manuscript. We do not allow these references, as the PLOS data access policy requires that all data be either published with the manuscript or made available in a publicly accessible database. Please amend the supplementary material to include the referenced data or remove the references.

Reviewers' comments:

Reviewer's Responses to Questions

**Comments to the Author**

1. Does this manuscript meet PLOS Global Public Health’s publication criteria? Is the manuscript technically sound, and do the data support the conclusions? The manuscript must describe methodologically and ethically rigorous research with conclusions that are appropriately drawn based on the data presented.

Reviewer #1: Yes

Reviewer #2: Yes

2. Has the statistical analysis been performed appropriately and rigorously?

Reviewer #1: N/A

Reviewer #2: N/A

3. Have the authors made all data underlying the findings in their manuscript fully available (please refer to the Data Availability Statement at the start of the manuscript PDF file)?

Reviewer #1: Yes

Reviewer #2: No

4. Is the manuscript presented in an intelligible fashion and written in standard English?

Reviewer #1: Yes

Reviewer #2: Yes

5. Review Comments to the Author

Reviewer #1: This is a very interesting study on using art based methods for collection of qualitative data on violence. The authors have ventured to understand the feasibility and experiences of the interviewers and the acceptability of this method in engaging participants to collect their information on experience of violence. What was most impressive about the paper was the strong grounding in a theoretical explanation of art based data collection methods. the authors have done a great job of explain the details of the methods they used for collecting data.They have explained the efforts in training of the interviewers and sensitivity and reflexivity of the interviewers. the details in which the methods are explained helps the reader understand the scientific rigour in the research. The findings are explained well with appropriate quotes. The discussion is elaborate and appropriate and well drawn out. I have some minor comments for improvement:

1. The main aim seems to be to understand feasibility and acceptability of art based interview methods in qualitative studies of violence. The results focus on the experiences of conducting these art based interviews.However, I would like to see some data specifically on feasibility - was the method more engaging? was the method easier than others? did the method involve less expenditure? was it more time efficient? what aspects of feasibility were analysed? I would also like to see some data on acceptability? were the participants more actively engaged and did they accept the method. Was there any instance where participants could not draw? was there reluctance to draw? such findings will enhance the understanding of the experience of using art based methods.

2. What measures were adopted to ensure credibility, transferability, dependability and confirmability of the qualitative findings? was some kind of triangulation of the findings possible? was it attempted?

3. How were the participants in the violence research sampled, how were the interviewers sampled for this study? was there some purposiveness in the sampling?

Reviewer #2: Thank you for this interesting paper which I enjoyed reading.

You provide a large number of references in support of the points you make - can you include in the reporting on what each of the various studies says, where the study was done/what the population was that the author(s) have engaged with? As you note yourself the context can influence how methods are used - so it would be helpful to provide the reader with the information on where each study was done. Is reference 64 a paper from the results of this pilot? If so, please say so, given this is a paper about methods it would be helpful for the reader to know where the study findings can be found.

In the first paragraph you mention the utility of the method for violence-related research - but you don't actually give any context for why you are interested in this. Can you provide a sentence background as to why this was your research area?

Line 92 - should the term `emotionally loaded events' be in quotes since it seems to be taken from reference 17.

Line 99 following you include a very useful piece on the interviewer experience - here you mention the novelty of papers and pencils being available in low resource settings. But this is not just a matter for the interviewers to manage - it also affects the respondent's experience. We know nothing about the socio-economic profile of the participants - can you say more about their background, the place, and the type of education they will have had - and how well that may have prepared them for research using novel methods?

Line 211 - what are quantifiable qualitative data? Information contained in a narrative that you count? I found this information on the quantitative interview rather confusing - perhaps you could insert a box showing a sequence of methods used in the interview?

Line 279ff -- did adults and children provide written consent and assent for the interviews? You only mention written consent for the FGDs.

The results section would benefit from some substantial reworking. You are over reliant on quotes at the moment and depend on the reader following the quotes and being able to draw from each of the quotes the meaning you wish to highlight. Please help the reader more - shift the balance so you paraphrase/ provide interpretative text much more, so the balance is shifted to have short, key quotes with the bulk of the text being your own words, to carry the reader through your arguments. The discussion is very long - and some of the material in that may work better in the results section, as you offer explanatory text around what an interviewee or the interviewer says about the experience of the methods..

The reference list requires some attention - there are some references with titles wholly in upper case, for example.

6. PLOS authors have the option to publish the peer review history of their article (what does this mean?). If published, this will include your full peer review and any attached files.

**Do you want your identity to be public for this peer review?** For information about this choice, including consent withdrawal, please see our Privacy Policy.

Reviewer #1: **Yes: **Vijayaprasad Gopichandran

Reviewer #2: No

---

## [Editor Report · Decision Letter 1]

1 Sep 2023

‘Through the drawings…they are able to tell you straight’ : using arts-based methods in violence research in South Africa

PGPH-D-23-00851R1

Dear Dr Meinck,

We are pleased to inform you that your manuscript '‘Through the drawings…they are able to tell you straight’ : using arts-based methods in violence research in South Africa' has been provisionally accepted for publication in PLOS Global Public Health.

Best regards,

Monica Malta

Academic Editor

This is an innovative and well-conducted study. Congratulations!